# Effects of Rearing System and Fattening Intensity on the Chemical Composition, Physicochemical Properties and Sensory Attributes of Meat from Young Crossbred (Holstein-Friesian × Hereford) Bulls

**DOI:** 10.3390/ani12070933

**Published:** 2022-04-05

**Authors:** Zenon Nogalski, Paulina Pogorzelska-Przybyłek, Monika Sobczuk-Szul, Monika Modzelewska-Kapituła

**Affiliations:** 1Department of Cattle Breeding and Milk Evaluation, Faculty of Animal Bioengineering, University of Warmia and Mazury in Olsztyn, Oczapowskiego 5, 10-719 Olsztyn, Poland; paulina.pogorzelska@uwm.edu.pl (P.P.-P.); monika.sobczuk@uwm.edu.pl (M.S.-S.); 2Department of Meat Technology and Chemistry, Faculty of Food Sciences, University of Warmia and Mazury in Olsztyn, Pl. Cieszyński 1, 10-719 Olsztyn, Poland; monika.modzelewska@uwm.edu.pl

**Keywords:** beef, nurse cow, meat quality, fatty acid composition, tenderness

## Abstract

**Simple Summary:**

Calves could be reared using automatic computer-controlled feeding machines or in a foster cow system, where one cow usually nurses two calves. The unrestricted access of calves to fresh milk in the foster cow system might have a beneficial effect on their health and later fattening performance and, ultimately, meat quality. Therefore, the aim of this study was to compare the effects of these rearing systems and fattening intensities on bioactive compounds content in beef originated from young bulls. An analysis of different rearing systems and fattening intensity levels indicates that natural calf rearing should be followed by intensive fattening to produce beef with optimal sensory attributes and a high nutritional value.

**Abstract:**

The study was aimed at determining the effects of the rearing system and intensity of fattening on beef physicochemical properties and sensory quality, fatty acid composition, and mineral compounds and vitamins concentration. The study was conducted using meat from 38 young, crossbred bull calves, which were reared with nurse cows (C) or were fed milk replacer (R). In the study, intensive (Int) or semi-intensive (SInt) fattening system were applied. The bulls were slaughtered at the age of 560 days and samples of the *longissimus lumborum* (LL) muscle were collected. Meat from C bulls was juicier (*p* < 0.05) and had a higher concentration of conjugated linoleic acid (CLA), eicosapentaenoic acid (EPA), and docosahexaenoic acid (DHA), as well as zinc (Zn), iron (Fe), and α-tocopherol, compared with meat from R bulls. The Int system resulted in the intramuscular fat (IMF) content increase (*p* < 0.01) and reduced shear force (*p* < 0.05), compared with the SInt system. Meat from Int bulls had a better eating quality and a higher monounsaturated fatty acid (MUFAs), Zn, and Fe; however, it had a lower proportion of polyunsaturated fatty acids (PUFAs) and α-tocopherol concentration than meat obtained from SInt bulls.

## 1. Introduction

In many countries where cattle are raised mostly for milk, to inseminate a part of cows and heifers in dairy herds, a beef bulls’ semen is usually used [1,2]. Several-day-old, crossbred calves being the progeny of dairy cows (or heifers) and beef bulls are sold to specialized in cattle fattening farmers. On such farms, calves are most often reared using automatic computer-controlled calf feeding machines [3]. However, those machines are expensive and the group housing of calves is associated with an increased risk of disease transmission. Therefore, alternative calf rearing systems are being sought [4]. One of these alternatives is a foster cow system [5], where one cow usually nurses two calves, and unrestricted access to fresh milk affects beneficially in later fattening results and, ultimately, beef quality [6,7]. Reddy et al. [8] demonstrated that nutritional stimulation in the early life stages of calves, which involved the administration of readily digestible high-energy diets, might result in metabolic imprinting demonstrated in the carcass and meat quality modification, especially backfat thickness and intramuscular fat scores.

The bases for beef production are male young bulls or steers, which are grazed intensively or semi-intensively, depending on the technology adopted [2]. In the semi-intensive system, animals are fattened using a fodder with a high proportion of silage and a small addition of concentrates, which extends the fattening time. This prolonging fattening period increases the animals age at slaughter, which may result in deterioration of the sensory quality of beef, i.e., its tenderness [9]. When grazing bulls, better results are obtained by using intensive feeding [10]. Bull calves reach slaughter maturity early and achieve optimal final body weight, and thus burden the environment to a lesser extent [11]. Moreover, bulls are a better choice than steers due to a higher feed efficiency and larger growth [12]. When environmental issues are discussed, it seems that there is a trend towards sustainable produced beef, especially that which is ‘grass-fed’. However, as it was reported by McGee et al. [12], the intensively fattened bulls, which received grass silage and concentrate showed lower greenhouse gases emission in relation to live, carcass, and meat weight gain.

Experiments conducted in our research team focus on the combined impact of the rearing system and intensity of feeding on the growth rates, carcass value, and meat quality from young bulls. Results regarding the growth rates of Polish Holstein–Friesian × Hereford calves, fattening performance and carcass characteristics were described in detail by Nogalski et al. [10]. It was concluded that a suckling system with nurse cows is more beneficial than using a milk replacer distributed from automated stations. Those nursed calves were healthier and had a higher body weight gain, and therefore were more suitable for fattening than calves which received milk replacer. Crossbred Polish Holstein–Friesian × Hereford bulls fattening indices and carcass traits were positively affected by an increased concentrate share in the fodder. However, the study [10] did not comprise the evaluation of meat quality. Thus, this study was aimed at analyzing the rearing system and the intensity of fattening effects on the physicochemical and sensory attributes of the meat, fatty acid composition, and concentration of mineral compounds and vitamins in meat from young, crossbred bulls. A hypothesis that natural rearing and intensive fattening of bulls contribute to increased concentration of bioactive compounds in beef, making it more valuable from a nutritional perspective, was tested in this study.

## 2. Materials and Methods

### 2.1. Calves

The study was conducted using young, crossbred bulls (*n* = 38) produced by inseminating Polish Holstein–Friesian cows with Hereford bulls’ semen. Until 150 days of age, those calves were raised naturally with nurse cows (C) or were fed milk replacer (R). During the rearing period, C calves turned out to be healthier—they had a better survival rate and a higher average daily body weight gain (by 0.15 kg) than R calves. After a 30-day transition period, the animals were fattened intensively (Int) or semi-intensively (SInt). The feeding period lasted from 181st day to 560th day of age. All details about calf rearing and bull fattening were provided in a previous paper [10]. The authors of the study obtained an approval for conducting the experiment from the University of Warmia and Mazury in Olsztyn Ethics Committee for Animal Experimentation (Decision No. 121/2010).

### 2.2. Slaughter and Meat Sampling

After termination of the fattening period, young, crossbred bulls were transported to a slaughterhouse, where they rested in individual boxes for 15 to 20 h; unlimited access to water was provided. The slaughter procedure was conducted in accordance with European Commission Regulation [13]. No electrical stimulation was applied to the carcasses. After 48 h, the pH48 value was measured in the *longissimus thoracis* (LT) muscle, between the 10th and 11th thoracic vertebrae (HI 8314 pH-meter with FC 200 combined electrode; Hanna Instruments, Olsztyn, Poland) [14]. Before the determinations, the pH-meter was calibrated with the use of pH 7 and pH 4 buffers. Ninety-six hours after the slaughter, the carcasses were dissected and approx. 1000 g samples of the *longissimus lumborum* (LL) muscle were collected from the right half-carcass of each animal (from the 1st to 3rd lumbar vertebrae). The muscles were transported to a laboratory under refrigerated conditions (delivery time of approx. 1 h) and kept at refrigerated temperature (4 ± 1 °C) overnight. Next, the muscles were divided and subjected to analyses.

### 2.3. Determination of Physicochemical Attributes of Meat

Prior to analyses, beef muscles were trimmed from external fat and individually comminuted using a meat grinder (ZMM4080, Zelmer S.A., Rzeszów, Poland) with 3 mm mesh. Dry matter was determined by drying meat samples at 103 ± 2 °C (UF55, Memmert GmbH+Co. KG, Schwabch, Germany) to a constant weight. The concentration of protein was determined according to the Kjeldahl method [15], using Büchi Labortechnik AG (Flavil, Switzerland) equipment; fat was determined according to the Soxhlet method using the Buechi B-811 extraction system, with hexane as a solvent [16] in oven-dried samples, and ash content was determined by ashing samples in quartz crucibles in a muffle furnace (FCF 22SM, Czylok, Poland) for 16h [17]. Drip loss in raw whole meat according to the Honikel method [18] and cooking loss—assessed on whole meat cooked to 75 °C [18]—were determined using a AKA 2200 (AXIS, Gdańsk, Poland) scale; drip loss and cooking loss indicated the water holding capacity (WHC) of muscle tissue.

LL color was evaluated in the CIE L*a*b* system [19] on the surface of freshly cut meat after 60 min of blooming using the Konica Minolta CR-400 (Sensing Inc., Osaka, Japan) (with a 2° view angle, D65 illuminant). Measurements were conducted in triplicate at randomly selected points and lightness (L*), redness (a*), and yellowness (b*) values were recorded. Chroma (C*) was calculated according to the formula C* = (a*^2^ + b*^2^)^0.5^, whereas hue angle was calculated according to the formula h° = atan (b*/a*)·180°/(Π).

A 2.5 cm thick steak (approx. 200g) obtained from each LL muscle was cooked in an individual plastic pouch in a water bath (W415E, Laboplay, Bytom, Poland) at 80 °C until a temperature of 75 °C was reached to determine the Warner–Bratzler shear force (WBSF, N). WBSF was measured on cuboid samples (10 × 10 mm, approx. 40 mm long, *n* = 5 from each steak) with the use of the Instron 5942 universal testing machine (Instron, Norwood, MA, USA) equipped with a V-shaped shear blade with a triangular aperture of 60° (load 500 N, head speed 200 mm/min). All details of the procedure were provided in [14].

### 2.4. Sensory Evaluation

Samples for sensory evaluation were prepared by cooking a 2.5 cm thick steak (approx. 200 g) obtained from each LL muscle in an individual plastic pouch in a water bath at 80 °C until an internal temperature of 75 °C was reached. Immediately after the termination of the thermal treatment, the samples were evaluated in accordance with Standard PN-ISO 4121 [20], by a six-person team trained and experienced in sensory evaluation of meat. A detailed description of the method was provided in Modzelewska-Kapituła et al. [14]. A total of seven sensory analysis sessions were performed, a maximum of six meat samples being assessed per session; the same panelists took part in all sessions. Panelists evaluated each LL sample in terms of typical beef aroma intensity (1, imperceptible; 5, extremely intense), juiciness (1, extremely dry; 5, extremely juicy), tenderness (1, extremely tough; 5, extremely tender), and typical beef taste intensity (1, imperceptible; 5, extremely intense).

### 2.5. Determination of Fatty Acid Composition, Mineral Compounds, and Vitamin A and E Content

Fatty acid composition was determined according to PN-EN ISO 5508 [21] and PN-EN ISO 5509 [22] standards. To obtain fatty acid methyl esters, the modified Peisker method [23] was used. The fatty acids were determined by gas chromatography, using the Varian CP 3800 system (Varian, Palo Alto, CA, USA). All the details of the analysis were provided in the paper by Nogalski et al. [24]. Saturated fatty acids (SFAs), unsaturated fatty acids (UFAs)—including monounsaturated fatty acids (MUFAs)—and polyunsaturated fatty acids (PUFAs) were reported as the relative percentage of total fatty acids along with the following calculated ratios: UFA/SFA, PUFA/SFA, and *n*-6/*n*-3 PUFA.

The minerals such as potassium, sodium, magnesium, zinc, and iron were determined using an atomic absorption spectrometer (Candela, Warsaw, Poland) according to the method described in Nogalski et al. [24].

The content of vitamins A (retinol) and E (α-tocopherol) was determined based on the applicable standards [25], with slight modifications [21], using a high-performance liquid chromatography (HPLC) using chromatograph 920-LC (Varian, Palo Alto, CA, USA), equipped with a Polaris C18-A (Agilent Technologies, Santa Clara, CA, USA) silica column). The separated compounds were identified by two detectors in tandem (UV-visible photodiode array detector and fluorescence detector). The analysis was conducted in duplicate.

### 2.6. Data Analysis

Data were analyzed using Statistica 13.3 software [26]. To determine if the rearing system (C and R) and fattening intensity (Int and SInt) affected meat attributes, the least squares method was used. To conduct the analysis, the following model was created:Y_ijk_ = μ + A_i_ + B_j_ + (AB)_ij_ + e_ijk_
where Y_ijk_ is the analyzed parameter value, μ is population mean, A_i_ is the effect of rearing system (1, 2), B_j_ is the effect of fattening intensity (1, 2), (AB)_ij_ is the rearing system x fattening intensity interaction, and e_ijk_ is random error.

To study the similarities between the treatments obtained from the nurse cow (C) and milk replacer (R) rearing systems, fattened intensively (Int) or semi-intensively (SInt), a cluster analysis was conducted using only these variables, which were affected by either rearing or fattening system. The data set included: fat content, WBSF, the proportion of conjugated linoleic acid (CLA), eicosapentaenoic acid (EPA), docosapentaenoic acid (DPA), docosahexaenoic acid (DHA), PUFAs, *n*-3, *n*-6/*n*-3 ratio, and Fe, Zn, and α-tocopherol contents. In the analysis, a single linkage between treatments and Euclidean distance were applied.

## 3. Results

### 3.1. The Influence of Rearing System and Intensity of Fattening on the Proximate Composition, Physicochemical, and Sensory Quality of Meat

Rearing system did not affect (*p* > 0.05) beef proximate composition (Table 1), nor its physicochemical properties (pH, color, WHC and WBSF, Table 2.) On the contrary, the intensity of feeding affected fat content in LL muscles, which was higher (*p* ≤ 0.01) in the Int system than in the SInt system (Table 1). However, no differences in the remaining components of meat tissue (dry matter, protein, and ash), as well as pH_48_ between Int and Sint raised bulls’ meat, were noted (*p* > 0.05).

The meat from Int bulls was lighter (L*), with a higher contribution of redness (a*) and yellowness (b*), but the noted differences were not significant (Table 2). This resulted in insignificant differences in chroma and hue angle between treatments. The experimental factors, such as rearing system and intensity of feeding, had no influence on the WHC of meat, which was expressed in the lack of significant differences in drip and cooking losses between the treatments (Table 2). The results of an instrumental evaluation of meat tenderness indicated that the meat of Int bulls was characterized by lower WBSF values than the meat of SInt bulls, and the effect of fattening intensity was significant (*p* ≤ 0.05) (Table 2).

Feeding levels also exerted a significant effect on the sensory attributes of the LL muscle (Table 3). Meat from Int bulls received higher scores, for all analyzed attributes, than meat from SInt animals. Only meat juiciness was affected by the method of calf rearing, and C bulls’ meat was juicier than R bulls meat (Table 3).

### 3.2. The Influence of Rearing System and Intensity of Fattening on the Longissimus Lumborum Fatty Acid Composition

The fatty acids concentrations in the IMF of the LL muscle were affected by both rearing system and fattening intensity (Table 4). The natural suckling system (C) and a better health status of calves contributed to an increase in conjugated linoleic acid (CLA), eicosapentaenoic acid (EPA), and docosahexaenoic acid (DHA) concentrations, whereas SInt fattening increased the concentrations of docosapentaenoic acid (DPA) and total PUFAs, while decreasing MUFAs proportion. Moreover, the natural suckling system (C) increased *n*-3 fatty acids proportion (*p* ≤ 0.05), thus decreasing the *n*-6/*n*-3 ratio (*p* ≤ 0.01).

### 3.3. The Influence of Rearing System and Intensity of Fattening on the Content of Macronutrients, Micronutrients, and Vitamins in Meat

Calf rearing system significantly affected (*p* < 0.05) the content of the analyzed micronutrients in the *longissimus lumborum* (LL) muscle (Table 5). C bulls’ meat showed significantly higher concentration of zinc (Zn) and iron (Fe) than meat from R bulls. The natural suckling system (C) also contributed to an increase in α-tocopherol concentration in meat. Meat from C bulls had higher (*p* ≤ 0.05) vitamin E content than meat from MF bulls. The concentrations of α-tocopherol, Zn, and Fe in meat were affected by fattening intensity. Beef from the SInt system showed a higher (*p* ≤ 0.05) vitamin E concentration compared with beef from the Int system, whereas Zn and Fe concentrations were higher in the meat from Int bulls.

### 3.4. Cluster Analysis Results

The cluster analysis clearly showed the two clusters differed with fattening intensity: animals from intensive fattening differed from those fattened semi-intensively (Figure 1). This suggests that although rearing system is important and affected some of the quality attributed to the meat, the fattening system strongly affects beef quality, including fat content and the proportion of valuable fatty acids, minerals such as Fe and Zn, and vitamin E content.

## 4. Discussion

### 4.1. The Proximate Composition and Physicochemical and Sensory Characteristics of the Longissimus Lumborum

In this study, the intensive fattening system (Int) resulted in a higher IMF content in the LL muscle compared with semi-intensive system (SInt). The results of this study are similar to those reported by Mezgebo et al. [27], who also found that using high levels of ground grain in cattle fodder increased the IMF content in beef. A similar was noted by Therkildsen et al. [28]. The IMF content at the beginning of the growth period in calves is likely to play a crucial role in shaping the IMF content after finishing. The phenomenon can be explained by the fact that adipose tissue starts to accumulate fat in the early weaning period [29] and the higher IMF content in this life stage, the higher level of IMF at the end of fattening and the same in meat. The results of this experiment are also consistent with those reported by Modzelewska-Kapituła and Nogalski [30], who noted that although intensity of feeding had an impact on the IMF content, dry matter and protein content remained unaffected in the *infraspinatus* muscles of Polish Holstein–Friesian bulls. On the other hand, Schoonmaker et al. [31] found that the IMF content increased in the muscles of early-weaned steers when they were fed *ad libitum* with rations containing a high proportion of concentrate during the growing stage. Nevertheless, the effect did not occur when steers were offered the same diet in the finishing period and the rates of fat deposition were slower. In this study, no effect of calf rearing system on the IMF content in LL muscle was noted, which is in agreement with the results reported by Greenwood et al. [32]. In their study, no impact of rearing system and BW gain on the IMF content in beef was shown.

The values of WBSF, which indicate the tenderness of muscle tissue, depend on the structure of both intramuscular connective tissue and myofibrillar proteins, which in turn are affected by muscle type and feeding system [14]. It was shown that animal feeding is a key factor which affects the biological reactions which proceed in muscle cells, such as muscle protein turnover [33]. In this study, the meat of Int bulls was characterized by lower WBSF values, compared with bulls reared in the SInt system. These results are different from those reported by Therkildsen et al. [28] and Cox et al. [34]. In the study by Therkildsen et al. [28], an increase in feed energy value had no influence on the WBSF values or tenderness of young bulls’ meat. Similarly, Cox et al. [34] reported that diet (forage vs. grains) had no effect on the loin WBSF values.

Intramuscular fat (IMF) indicates the quantity of fat in muscles and is visible, described as fat spots or deposits (so called marbling) in some beef carcass elements. IMF content positively influences the sensory attributes of meat, such as juiciness, flavor, and tenderness [14,29,35], in addition to WBSF values [36]. This was also noted in the present study, where the higher IMF concentration in Int bulls’ meat corresponded with an increase in the desirable sensory properties of the meat. Moreover, meat with the best sensory quality (group Int) had a lower value of WBSF. Meat from C bulls was juicier, which could result from a higher growth rate of nursed calves [10]. Bispo et al. [37] found that nursed calves were characterized by higher BW gains and their meat was juicier, compared with calves that could not suckle from their mothers. In contrast, Hennessy et al. [38] demonstrated that samples of the *longissimus lumborum et thoracis* muscle originated from calves showing lower growth rates before weaning had a better tenderness than those from animals which had higher growth rates. In this study, meat from C calves had a somewhat higher IMF content than meat from R calves (0.09% and 0.2%, a non-significant difference), which could partially explain the observed difference in meat juiciness.

### 4.2. The Quality of Fat in the Longissimus Lumborum Muscle

The natural suckling system and a better health status of calves [10] contributed to the elevated levels of value from human nutrition fatty acids such as CLA, EPA, and DHA. A high share of SFAs and a low proportion of PUFAs in the fat result from the hydrogenation of dietary fat by ruminal microbiota [39]. For optimal results, beef producers should decrease the concentrations of SFAs in fat and/or increase the content of PUFAs, especially *n*-3 fatty acids [40]. In this study, the natural suckling system caused an increase in the proportion of *n*-3 fatty acids. The findings of De la Fuente et al. [41] suggest that artificially reared lambs have a lower rumen functionality, which affects their absorption of fatty acids synthesized in the rumen via the anaerobic microbial fermentation of fiber and starch [42]. In a study by Osorio et al. [43], ewes’ milk had a beneficial influence on the concentrations of *n*-3 fatty acids and the *n*-6/*n*-3 PUFA ratio in the IMF of lamb meat, compared to milk replacer. Wielgosz et al. [4] reported that diseases and infections during the rearing period negatively affected IMF composition in calves. In addition, in this experiment, diseases that occurred in the pre-weaning period might have influenced the post-weaning health status of calves and, consequently, the fatty acid profile. In this study, SInt bulls’ meat had a lower IMF content, but the composition of the fat was more favorable compared to that from Int bulls, due to the fact that semi-intensive fattening increased DPA and total PUFAs contents in meat.

### 4.3. The Content of Macronutrients, Micronutrients, and Vitamins in Meat

As a result of natural rearing, Zn and Fe contents increased in meat, most likely because calves could obtain high-quality milk by suckling from nurse cows. Bispo et al. [6] reported that the earliest weaned calves produced meat with a lower nutritional value. The contents of minerals in beef may be associated with the fat content, which was shown in a study Williams et al. [44], where Zn, Fe, P, Na, and K concentrations were negatively correlated to carcass fatness [44]. A similar result was noted in this study, where higher Zn and Fe contents were noted in the meat with a lower IMF content. From a nutritional perspective, the meat from C bulls was a rich source of Zn, satisfying 52% and 23% of the RDA (Recommended Dietary Allowance [45]) in adult females and males, respectively, when consumed at the amount of 100 g (the weight of cooked meat).

In this study, both the natural suckling system and semi-intensive fattening had a beneficial influence on the α-tocopherol concentration in meat. This can be explained by the fact that in the meat of animals fed mainly roughage, which is used in semi-intensive fattening, higher concentrations of vitamin E are noted [46]. At the same time, the roughage feeding also causes elevated proportions of long-chain unsaturated fatty acids (DPA and total PUFAs). It should be noted that, in the meat which has an increased content of PUFAs, there is a higher risk of detrimental lipid oxidation and off-flavor development during shelf-life. However, the lipid oxidation in meat might be inhibited by high vitamin E content [47]. Therefore, the simultaneous increase in PUFA and vitamin E is highly desirable in light of beef shelf-life. Antioxidants, such as vitamin E, not only prevent those undesirable changes in lipids, but slow down metmyoglobin formation, in addition to the same unfavorable changes in meat color [47]. To exert the inhibition, a minimum of 3–4 µg of α-tocopherol/g of fresh beef is needed [48]. In our study, the concentration of α-tocopherol was within this range, excluding beef from bulls fed with milk replacer fattened intensively; therefore, a positive effect on lipid oxidation and color stability might be expected. However, to verify this hypothesis, further studies are needed.

## 5. Conclusions

In conclusion, both rearing and feeding systems affected beef quality; however, the influence of feeding system was stronger than rearing system. The advantage of suckling system over milk replacer was shown in this study; therefore, it is recommended to raise calves using less valuable cows for nursing. Meat from bulls raised using nurse cows, compared with meat from bulls, which received milk replacer, was juicier and the concentration of functional fatty acids (CLA, total *n*-3 PUFA, including EPA and DHA), in addition to Zn, Fe, and α-tocopherol, was higher. Intensive fattening, compared with semi-intensive fattening, contributed to the IMF increase in the LL muscle and a reduction in the values of WBSF. Meat from intensively reared bulls was more desirable in terms of sensory quality, and had higher concentrations of MUFAs, Zn, and Fe. Although the beef obtained from semi-intensive fattening had more desirable fatty acid composition in terms of PUFAs and DPA proportions, it had also a lower intramuscular fat concentration; therefore, it will not probably be a much better source of those components in a human diet, compared with the meat from intensive fattening. An analysis of different rearing systems and fattening intensity levels indicates that natural calf rearing should be followed by intensive fattening to produce beef with optimal sensory attributes, delivering health benefits to consumers.

## Figures and Tables

**Figure 1 animals-12-00933-f001:**
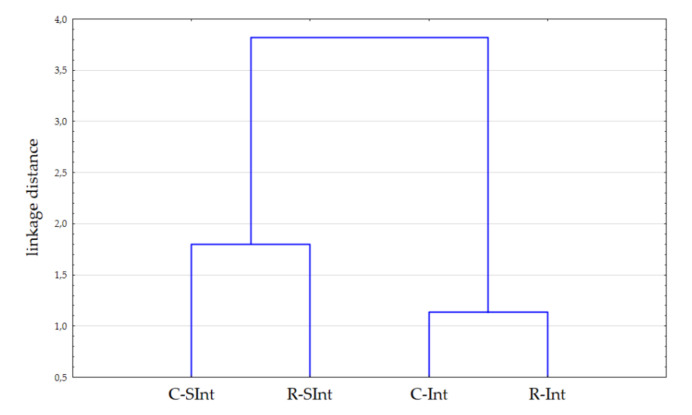
Complete-linkage dendogram for bulls obtained from nurse cow (C) and milk replacer (R) rearing systems fattened intensively (Int) or semi-intensively (SInt) based on fat content, WBSF, the proportion of conjugated linoleic acid (CLA), eicosapentaenoic acid (EPA), docosapentaenoic acid (DPA), docosahexaenoic acid (DHA), PUFAs, and *n*-3, *n*-6/*n*-3 ratio, Fe, Zn, α-tocopherol contents.

**Table 1 animals-12-00933-t001:** The proximate composition of the *longissimus lumborum* muscle.

Specification	R		C		SEM	*p*-Value		
	Int	SInt	Int	SInt		RS	FS	RSxFS
Dry matter [%]	26.2	25.53	25.71	25.1	0.214	0.287	0.139	0.949
Protein [%]	21.92	21.83	21.86	22.04	0.110	0.714	0.827	0.532
Fat [%]	2.92	1.74	3.03	1.94	0.280	0.477	0.002	0.933
Ash [%]	1.08	1.07	1.08	1.08	0.005	0.775	0.531	0.689

RS—rearing system: R—milk replacer or C—nurse cow; FS—fattening system: Int—intensive fattening; SInt—semi-intensive fattening; SEM—standard error of the mean.

**Table 2 animals-12-00933-t002:** Physicochemical properties of the *longissimus lumborum* muscle.

Specification	R		C		SEM	*p*-Value		
	Int	SInt	Int	SInt		RS	FS	RSxFS
pH48	5.58	5.59	5.52	5.52	0.033	0.331	0.876	0.963
L*	36.57	35.91	35.94	35.23	0.536	0.561	0.981	0.547
a*	18.75	18.43	18.22	17.56	0.390	0.395	0.545	0.830
b*	14.12	13.80	16.69	13.68	0.877	0.499	0.458	0.758
C*	23.47	23.02	24.23	22.78	0.456	0.432	0.326	0.636
h°	36.98	36.82	42.49	37.92	0.832	0.151	0.065	0.412
Drip loss [%]	1.84	2.37	2.18	1.95	0.184	0.913	0.701	0.324
Cooking loss [%]	33.72	32.65	34.04	34.73	0.555	0.297	0.868	0.442
WBSF [N]	38.87	42.3	38.78	43.79	2.332	0.135	0.047	0.242

RS—rearing system: R—milk replacer or C—nurse cow; FS—fattening system: Int—intensive fattening; SInt—semi-intensive fattening; SEM—standard error of the mean; L*—lightness; a*—redness; b*—yellowness; C*—chroma; h°—hue angle; WBSF—Warner-Bratzler shear force; WHC—water-holding capacity (drip loss and cooking loss).

**Table 3 animals-12-00933-t003:** Sensory assessment results of the *longissimus lumborum* muscle.

Specification	R		C		SEM	*p*-Value		
	Int	SInt	Int	SInt		RS	FS	RSxFS
Aroma	4.7	4.2	4.7	4.5	0.067	0.260	0.023	0.503
Tenderness	4.3	3.8	4.5	3.5	0.113	0.879	0.001	0.235
Juiciness	4.0	3.6	4.3	3.8	0.071	0.045	0.000	0.278
Taste	4.6	4.1	4.6	4.2	0.067	0.948	0.001	0.361

RS—rearing system: R—milk replacer or C—nurse cow; FS—fattening system: Int—intensive fattening; SInt—semi-intensive fattening; SEM—standard error of the mean; scale used in the evaluation from 1 to 5 points.

**Table 4 animals-12-00933-t004:** The influence of rearing system and intensity of fattening on the fatty acids proportions (% of total) in the *longissimus lumborum* intramuscular fat.

Specification	R		C		SEM	*p*-Value		
	Int	SInt	Int	SInt		RS	FS	RSxFS
CLA	0.244	0.317	0.344	0.372	0.014	0.045	0.656	0.666
C 20:5 EPA	0.104	0.109	0.122	0.173	0.013	0.033	0.967	0.911
C 22:5 DPA	0.214	0.292	0.216	0.393	0.027	0.181	0.047	0.422
C 22:6 DHA	0.032	0.062	0.073	0.103	0.015	0.031	0.309	0.303
SFAs	48.65	51.20	48.37	49.80	0.456	0.579	0.114	0.331
UFAs	51.34	48.80	51.62	50.21	0.456	0.579	0.225	0.231
MUFAs	47.04	43.38	47.13	44.49	0.613	0.694	0.044	0.150
PUFAs	4.30	5.41	4.49	5.72	0.381	0.783	0.038	0.841
PUFA/SFA	0.109	0.106	0.109	0.122	0.008	0.932	0.372	0.632
MUFA/SFA	0.95	0.85	0.89	0.94	0.020	0.660	0.502	0.074
*n*-6	3.60	3.67	3.41	3.68	0.289	0.889	0.778	0.867
*n*-3	1.40	1.42	1.81	1.79	0.077	0.021	0.787	0.933
*n*-6/*n*-3	2.51	2.54	1.75	1.99	0.097	0.002	0.432	0.517

RS—rearing system: R—milk replacer or C—nurse cow; FS—fattening system: Int—intensive fattening; SInt—semi-intensive fattening; SEM—standard error of the mean; CLA—conjugated linoleic acid; EPA—eicosapentaenoic acid; DPA—docosapentaenoic acid; DHA—docosahexaenoic acid; SFAs—saturated fatty acids; UFAs—unsaturated fatty acids; MUFAs—monounsaturated fatty acids; PUFAs—polyunsaturated fatty acids.

**Table 5 animals-12-00933-t005:** The influence of rearing system and intensity of fattening on the content (mg/100 g of fresh meat) of minerals and vitamins.

Minerals/Vitamins	R		C		SEM	*p*-Value		
	Int	SInt	Int	SInt		RS	FS	RSxFS
K	501.1	496.4	482.8	500.9	3.65	0.346	0.354	0.122
Na	59.6	65.1	60.3	58.2	1.50	0.313	0.589	0.209
Mg	21.7	20.5	20.6	20.4	0.44	0.539	0.433	0.617
Zn	4.1	3.8	4.7	4.3	0.09	0.042	0.033	0.228
Fe	1.7	1.4	2.0	1.8	0.06	0.034	0.041	0.879
Retinol	0.081	0.086	0.090	0.067	0.002	0.808	0.691	0.511
α-tocopherol	0.202	0.321	0.355	0.431	0.062	0.034	0.045	0.612

RS—rearing system: R—milk replacer or C—nurse cow; FS—fattening system: Int—intensive fattening; SInt—semi-intensive fattening; SEM—standard error of the mean.

## Data Availability

The data presented in this study are available on request from the corresponding author.

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
