# Peer review of "Effects of Rearing System and Fattening Intensity on the Chemical Composition, Physicochemical Properties and Sensory Attributes of Meat from Young Crossbred (Holstein-Friesian × Hereford) Bulls"

_animals, 2022, doi:10.3390/ani12070933_

Round 1

Reviewer 1 Report

Comments on the paper by Nogalski and co-workers submitted for publication in Animals journal. I read this nice and short paper with great interest. It aimed to determine the effects of rearing system and intensity of fattening on beef physicochemical properties and sensory quality, fatty acid composition and mineral compounds and vitamins concentration. The authors used 38 young cross-bred bull calves, which were reared with nurse cows (NC) or were fed milk replacer (MF). The animals were fattened intensively (I) or semi-intensively (SI). The bulls were slaughtered at a similar age and the striploin was used for the characterization.

The methods were well described.

I can see that from this study, a multivariate analysis such as principal component analysis is missing to better see the separation or no among the group. I highly encourage the authors to consider such analysis that will further strengthen the quality of the work.

The introduction is short, and it needs a second paragraph that can be focused on young bulls only, with the potential role they can play on farming sustainability.

Hue angle is missing in this article. Can you argue why it is not included?

Table 1 can be separated into two tables. The quality should be in a separate table.

Line 216 is the reference 27 about cattle? Check and amend as necessary.

Line 238, this references is recent and accurate to cite https://doi.org/10.1016/j.livsci.2018.06.011

Few references on young bulls in the topic of this paper can be also added from the recent literature.

Author Response

Reviewer 1

General comment: I read this nice and short paper with great interest. It aimed to determine the effects of rearing system and intensity of fattening on beef physicochemical properties and sensory quality, fatty acid composition and mineral compounds and vitamins concentration. The authors used 38 young cross-bred bull calves, which were reared with nurse cows (NC) or were fed milk replacer (MF). The animals were fattened intensively (I) or semi-intensively (SI). The bulls were slaughtered at a similar age and the striploin was used for the characterization. The methods were well described.

A: Thank you!

C1. I can see that from this study, a multivariate analysis such as principal component analysis is missing to better see the separation or no among the group. I highly encourage the authors to consider such analysis that will further strengthen the quality of the work.

A: we try to apply PCA but the results of the analysis were not satisfactory and therefore a cluster analysis was applied, which show a similarities between treatments. The figure with a result of cluster analysis was inserted into a paper.

C2. The introduction is short, and it needs a second paragraph that can be focused on young bulls only, with the potential role they can play on farming sustainability.

A: A part of Introduction describing a sustainability of rearing bull has been added.

C3. Hue angle is missing in this article. Can you argue why it is not included?

A: Hue angle was calculated and values were shown in a Table.

C4. Table 1 can be separated into two tables. The quality should be in a separate table.

A: According to the comment, the table was divided and in the current version separate tables for composition and physicochemical attributes are presented.

C5. Line 216 is the reference 27 about cattle? Check and amend as necessary.

A: Indeed, the reference is not about cattle. It was replaced by the one referring to bulls (Modzelewska-KapituÅ‚a, M.; Nogalski, Z. The influence of diet on collagen content and quality attributes of infraspinatus muscle from Holstein–Friesian young bulls. Meat Sci. 2016, 117, 158-162. https://doi.org/10.1016/j.meatsci.2016.03.003)

C: Line 238, this references is recent and accurate to cite https://doi.org/10.1016/j.livsci.2018.06.011

Few references on young bulls in the topic of this paper can be also added from the recent literature.

A: The reference has been added.

Reviewer 2 Report

Authors should consider the following recommendations

Line 2: delete the word Title

Line 2: The title does not reflect the complete content of the work, although it considers the effect of the rearing system and fattening intensity on the content of bioactive compounds, the title is not considering the chemical composition, physicochemical properties and sensory attributes.

Line 84-87: indicate the reference of the method used for the determination of pH

Line 95: sometimes it is used in the word physico-chemical or physicochemical, it is necessary to standardize it in the document

Line 96: insert information about the brand, model and country of the equipment used (meat grinder)

Line 97: insert information about the brand, model and country of the equipment used for the determination of dry matter (drying oven)

Line 97: sometimes uses °C and ºC, it is necessary to homogenize in the document

Line 98: insert information about the brand, model and country of the equipment used for the determination of protein, fat and ash, as well as the name of the technique/method for its determination

Lines 99,100: indicate the name of the methods used to determine the cooking loss and water holding capacity, as well as information on the equipment used

Line 102: change min. of by min of

Line 103: explain why use 2° view angle instead of 10°

Line 105: the values of h* (hue) were not considered?

Line 107: insert information about the brand, model and country of the equipment used (water bath)

Line 137: insert information about the brand, model and country of the equipment used

Line 148: 3.1. The influence of rearing system and intensity of fattening on the proximate composition, 148 color, pH, WBSF, WHC and sensory quality of meat or/ 3.1. The influence of rearing system and intensity of fattening on the proximate composition, physicochemical and sensory quality of meat

Line 150: no information is inserted on the results regarding the other components of the meat samples evaluated

Line 151: Table 1, according to the place inserted, gives the impression that it only talks about the fat content

Line 153: no information is inserted on the results regarding the drip loss and cooking loss of the meat samples evaluated

Line 161: no information about WHC is inserted in Table 1

Line 164: C* - chroma; WBSF - Warner…; WHC - water…

Line 276: If lipid oxidation in meat might be inhibited by high vitamin E contents, I consider it important to evaluate the lipid oxidation state of the samples used in this study, either by evaluating substances reactive to thiobarbituric acid. This in order to establish if the increase in this bioactive compound (α-tocopherol) is really slowing down the oxidation of the samples.

Line 293: in the conclusion the findings are not presented in the order in which the results were presented, in addition, certain results obtained are omitted

Line 326:… 337–342

Line 344: … 1425–1434

Line 369: … 25–33

Line 371: scientific names should be written in italic text format

Line 388: 843–855

Line 388: the doi of the reference was not included

Line 403: delete spaces…715–724

Line 405: 73–79

Line 409: 75–83

Line 409: the doi of the reference was not included

Line 414: 127–134

Line 420: 384–394

Author Response

Reviewer 2

C1. Line 2: delete the word Title

A: deleted

C2. Line 2: The title does not reflect the complete content of the work, although it considers the effect of the rearing system and fattening intensity on the content of bioactive compounds, the title is not considering the chemical composition, physicochemical properties and sensory attributes.

A: The title was modified to cover all of parameters studied

C3. Line 84-87: indicate the reference of the method used for the determination of pH

A: The reference was provided

C4. Line 95: sometimes it is used in the word physico-chemical or physicochemical, it is necessary to standardize it in the document

A: corrected

C5. Line 96: insert information about the brand, model and country of the equipment used (meat grinder)

A: provided

C6. Line 97: insert information about the brand, model and country of the equipment used for the determination of dry matter (drying oven)

A: provided

C7. Line 97: sometimes uses °C and ºC, it is necessary to homogenize in the document

A: Corrected

C8. Line 98: insert information about the brand, model and country of the equipment used for the determination of protein, fat and ash, as well as the name of the technique/method for its determination

A: All the details have been provided

C9. Lines 99,100: indicate the name of the methods used to determine the cooking loss and water holding capacity, as well as information on the equipment used

A: the information was provided

C10. Line 102: change min. of by min of

A: Corrected

C11. Line 103: explain why use 2° view angle instead of 10°

A: the device used by us in the study operated only on 2° Observer.

C12. Line 105: the values of h* (hue) were not considered?

A: Hue angle was calculated and values were shown in a Table

C13 Line 107: insert information about the brand, model and country of the equipment used (water bath)

A: The information was provided

C14. Line 137: insert information about the brand, model and country of the equipment used

All the details have been provided

C.15. Line 148: 3.1. The influence of rearing system and intensity of fattening on the proximate composition, 148 color, pH, WBSF, WHC and sensory quality of meat or/ 3.1. The influence of rearing system and intensity of fattening on the proximate composition, physicochemical and sensory quality of meat

A: Thank you, the subtitle was modified

C.16 Line 150: no information is inserted on the results regarding the other components of the meat samples evaluated

A: The text has been revised and modified to mention all studied attributes.

C17. Line 151: Table 1, according to the place inserted, gives the impression that it only talks about the fat content

A: The placement of Table 1 was modified and it was placed just after the presentation of proximate composition results.

C18. Line 153: no information is inserted on the results regarding the drip loss and cooking loss of the meat samples evaluated

A: The information about the lack of differences in WHC (drip loss and cooking loss) have been provided in the revised version of the manuscript.

C19. Line 161: no information about WHC is inserted in Table 1

A: In this study, just as indicted in the materials and methods section, WHC was evaluated based on drip loss and cooking loss. The information about it was placed below Table 2 “WHC - water-holding capacity (drip loss and cooking loss).” In the table the value were provided and in the text WHC (drip and cooking losses) were described in the light of the effect of rearing system and feeding intensity.

C20. Line 164: C* - chroma; WBSF - Warner…; WHC - water…

A: corrected

C21. Line 276: If lipid oxidation in meat might be inhibited by high vitamin E contents, I consider it important to evaluate the lipid oxidation state of the samples used in this study, either by evaluating substances reactive to thiobarbituric acid. This in order to establish if the increase in this bioactive compound (α-tocopherol) is really slowing down the oxidation of the samples.

A: We totally agree with the Reviewer – to verify the effect of increased level of vitamin E on lipid oxidation an additional determination such as TBARS should be conducted. We are grateful for the comment and we will bare it in mind when designing future studies. The need for verifying the antioxidant effect of vitamin E was mentioned in the text.  Was indicated in the discussion section.  

C22. Line 293: in the conclusion the findings are not presented in the order in which the results were presented, in addition, certain results obtained are omitted

A: The conclusion section was modify and lacking information was added.

C23. Line 326:… 337–342

A: corrected

C24. Line 344: … 1425–1434

A: corrected

C25. Line 369: … 25–33

A: corrected

C26. Line 371: scientific names should be written in italic text format

A: corrected

C27. Line 388: 843–855

A: corrected

C28. Line 388: the doi of the reference was not included

A: doi number has been provided

C29. Line 403: delete spaces…715–724

A: corrected

C30. Line 405: 73–79

A: corrected

C31. Line 409: 75–83

A: corrected

C32. Line 409: the doi of the reference was not included

A: doi number has been provided

C33. Line 414: 127–134

A: corrected

C34. Line 420: 384–394

A: corrected

Reviewer 3 Report

The study compared two different rearing systems as well as two intensities of fattening process on meat quality of beef cattle. The study concept is clearly presented. The manuscript is well-written with nice explanations of the results. The authors provide a good introduction explaining the background and rationale for their study and their experimental objectives are clearly stated. I list below some comments for the authors’ consideration:

1) Line 52, “was investigated” should be deleted.

2) Line 58-64, these two sentences were repeating.

3) How did the meat sample prepare for the chemical analysis, oven-dried or lyophilization?

4) Line 135, authors’ name should be added.

5) Line 142, “model” is better than “formula”.

6) As described in Methods, fat content of the meat sample was determined as crude fat which was not representative for intramuscular fat.

7). Why marbling score analysis was not conducted which was more intuitive for IMF of meat.

8) Letter “P” represented for significance should be italic.

9) Line 178, MUFAs were decreased by SI fattening shown in Table 3.

10) Line 262-265, results related to diseases and infections’ effect on health and fatty acid were not presented.

11) Discussion on vitamins content in meat was a little bit confusion. Vitamin E was increased both by NC system and SI fattening, but author attributed this effect to higher roughages in the diet which natural suckling was not included. Moreover, why did author mention the level of vitamin E in the present study was not reach the minimum inhibition requirement.

12) Conclusion about PUFAs should be changed according to the results.

Author Response

Reviewer 3

General comment: The study compared two different rearing systems as well as two intensities of fattening process on meat quality of beef cattle. The study concept is clearly presented. The manuscript is well-written with nice explanations of the results. The authors provide a good introduction explaining the background and rationale for their study and their experimental objectives are clearly stated. I list below some comments for the authors’ consideration:

C1. Line 52, “was investigated” should be deleted.

A: deleted

C2. Line 58-64, these two sentences were repeating.

A: corrected; the unnecessary part was deleted

C3. How did the meat sample prepare for the chemical analysis, oven-dried or lyophilization?

A: Samples were oven-dried. This was indicated in the text. 

C4. Line 135, authors’ name should be added.

A: The author’s name was inserted

C5. Line 142, “model” is better than “formula”.

A: corrected

C6. As described in Methods, fat content of the meat sample was determined as crude fat which was not representative for intramuscular fat.

A: Fat content which was determined was intramuscular fat (line 125).

C7. Why marbling score analysis was not conducted which was more intuitive for IMF of meat.

A: Thank you for your suggestion, we will introduce this in future studies.

C8. Letter “P” represented for significance should be italic.

A: corrected

C9. Line 178, MUFAs were decreased by SI fattening shown in Table 3.

A: Thank you for the comment. It has been corrected.

C10. Line 262-265, results related to diseases and infections’ effect on health and fatty acid were not presented.

A: the information has been provided in the Material and methods section – the bulls from the suckling system had a better survival rate and a higher daily gains. We did not enclose it in the Discussion section to avoid repeating the information.

C11. Discussion on vitamins content in meat was a little bit confusion. Vitamin E was increased both by NC system and SI fattening, but author attributed this effect to higher roughages in the diet which natural suckling was not included. Moreover, why did author mention the level of vitamin E in the present study was not reach the minimum inhibition requirement.

A: Thank you very much for the comment. The discussion was corrected, and focused on semi-intensive feeding which increased vitamin content. The vitamin E contents shown in Table fits to the indicated limit and therefore this part of the discussion was corrected.

C12. Conclusion about PUFAs should be changed according to the results.

A: The conclusion section was modify and lacking information about PUFA was added.

Round 2

Reviewer 2 Report

The recommended observations were attended